fluid mechanics

rotor blades, swift, swept wings, particle image velocimetry

**Author for correspondence:**
Roi Gurka
e-mail: rgurka@coastal.edu

# Wake characteristics of a freely rotating bioinspired swept rotor blade

Asif Shahriar Nafi[1], Krishnamoorthy Krishnan[1], Anup K. Debnath[1], Erin E. Hackett[1] and Roi Gurka[2]

[1]Department of Coastal and Marine System Science, and [2]Department of Physics and Engineering Science, Coastal Carolina University, Conway, SC, USA

ASN, 0000-0002-4770-5249; AKD, 0000-0002-8139-2573; EEH, 0000-0002-1294-6501; RG, 0000-0002-8907-6663

Rotor blades can be found in many engineering applications, mainly associated with converting energy from fluids to work (or electricity). Rotor blade geometry is a key factor in the mechanical efficiency of the energy conversion process. For example, wind turbines' performance directly depends on the blade geometry and the wake flow formed behind them. We suggest to use a bioinspired blade based on the common swift wing. Common swift (*Apus apus*) is known to be a long-distance flyer, able to stay aloft for long periods of time by maintaining high lift and low drag. We study the near-wake flow characteristics of a freely rotating rotor with swept blades and its aerodynamic loads. These are compared with a straight-bladed rotor. The experiments were conducted in a water flume using particle image velocimetry (PIV) technique. Both blades were studied for four different flow speeds with freestream Reynolds numbers ranging from 23 000 to 41 000. Our results show that the near wake developed behind the swept-back blade was significantly different from the straight blade configuration. The near wake developed behind the swept-back blade exhibited relatively lower momentum loss and suppressed turbulent activity (mixing and production) compared with the straight blade. Comparing the aerodynamic characteristics, though the swept-back blade generated relatively less lift than the straight blade, the drag was relatively low. Thus, the swept-back blade produced two to three times higher lift-to-drag ratio than the straight blade. Based on these observations, we suggest that, with improved design optimizations, using the swept-back configuration in rotor blades (specifically used in wind turbines) can improve mechanical efficiency and reduce the energy loss during the conversion process.

# 1. Introduction

Over the past decade, there has been a growing number of studies to maximize the performance of wind turbines and their layouts for wind farms [1,2]. As the sites for wind farm development are not highly abundant in number, it is necessary to ensure maximum power production per unit area. However, clustering the wind turbines in wind farms is a daunting task, as they operate in the atmospheric boundary layer which is turbulent in nature. For example, the increased wake turbulence and velocity deficit generated by the upstream wind turbines that interact with the downstream wind turbines [3] may reduce the total power generation from a wind farm due to the nonlinear interaction between the wake flow structures. This interaction not only affects the performance of the downstream wind turbines but also increases dynamic loading on them that may affect the turbines' life expectancy by potentially causing component failures [4]. To better understand parameters involved in such interactions, it is important to study the characteristics of the wake behind rotor blades because they are an essential component in converting wind kinetic energy to mechanical work.

The velocity field in the wake of a rotating blade comprises two regions: a near-wake region and a far-wake region. The former is characterized by helical vortex structures and is influenced by the rotor aerodynamics, and the latter features a wake profile with a Gaussian velocity distribution and is influenced by the surrounding environment that prompts rapid wake expansion [5,6]. It is assumed that the near-wake region is between one to five rotor diameters ($D$) away from the rotor disc [7]. The far-wake region starts at the downstream location where the tip and hub vortices merge [8].

Over the past few decades, wind turbine research has mainly focused on predicting and optimizing power production of wind farms and individual wind turbines through modelling their wake (i.e. [9,10]) based on field and controlled laboratory experiments as well as numerical simulations [11]. Due to limited spatial and temporal resolution of *in situ* and remote flow field diagnostic techniques (e.g. LiDAR, radar), high-resolution data on turbulent flow fields, especially in the anisotropic near-wake region, are lacking [12]. The dynamics of the near wake can affect the far-wake development significantly [13]. To overcome such limitations of field studies, scaled-down models and kernel studies of specific wind turbine components are essential. Many wind tunnel experiments were conducted on scaled models of wind turbines to help validate and improve numerical models of wake flow behind wind turbines [5,14]. Flow field measurements using particle image velocimetry (PIV) have become increasingly popular in terms of wind turbine research as it allows researchers to measure and analyse two- or three-dimensional velocity vector fields around a wind turbine [15–17].

In general, the near-wake region is less studied than the far-wake region. Whale *et al*. [18] investigated the near-wake properties of a model wind turbine using PIV to compare with field data from two full-scale wind turbines. They showed some similarities in the wake characteristics sampled using PIV data when compared with the full-scale wind turbine located in flat terrain. Hirahara *et al*. [19] designed a micro wind turbine (defined as wind turbine whose rotor diameter is less than 1.25 m) and studied its wake characteristics as well as its performance in various urban conditions. They found the power coefficient of the wind turbine to be 0.40 when the tip speed ratio was 2.7 (a utility-scale wind turbine can achieve peak power coefficient of 0.44–0.48). Sherry *et al*. [20] studied the near-wake data of a model wind turbine in a water channel using PIV. They characterized the root and tip vortices in the near wake and studied the wake meandering phenomenon that is linked to root and tip vortex instabilities. Their results suggest that low velocity in the wake center region increases in area as the tip speed ratio increases, but the pitch of the tip vortex gets smaller. They also found that wake meandering is suppressed by the increase in pitch of the tip vortex system until the wake becomes unstable. In order to investigate the effect of incoming boundary layer flow on the wake development of a model wind turbine, Zhang *et al*. [21] used stereoscopic PIV to study the near-wake flow and found significant turbulence enhancement at a distance of three rotor diameters downstream from the rotor plane. These studies characterize the near-wake region where interactions among turbine-generated coherent structures dominate, which significantly affects the far-wake development [22].

This study focuses on characterizing the near-wake flow of a bioinspired rotor blade. We are inspired by the common swift bird [23]. Common swifts (*Apus apus*) are known for their high-performance flight (lift-to-drag ratio equal to 12.5 for gliding and 7.7 for flapping). Their wing morphology and flight characteristics have been studied thoroughly [24–26]. Prior studies have shown it to use leading-edge vortices (LEV) during gliding [27,28]. When a stable LEV is formed over the wing of common swifts, it augments lift as well as causes delayed stall [29]. These are commonly known to form over linear (delta wing) [30,31] and more recently over nonlinear swept-back (swift-like) wing geometries [32,33]. Ashwill *et al*. [34] developed a 27 m long swept blade (Sweep-Twist Adaptive Rotor blade) to be

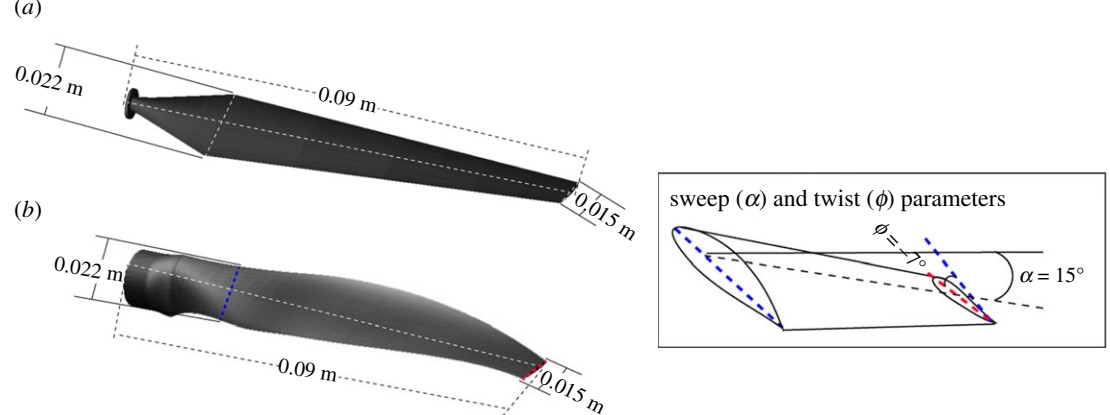

**Figure 1.** CAD model of rotor blades (*a*) straight blade (only twist parameter present) (*b*) swept blade (both twist and sweep parameters present).

operated at low wind speed which was field-tested in California, and the data showed 10–12% increase in wind energy capture as compared with the straight-bladed baseline model of the wind turbine. CFD simulation of a swept wind turbine blade was reported to provide a higher power coefficient than a straight blade [35]. Existing literatures on swept rotor blade mainly focus on the design concept and analysis of aerodynamic performance [36,37]. To the best of our knowledge, there has not been any literature addressing the near-wake characteristics of the swept blade wind turbine. The present study partially fills this knowledge gap by characterizing the near-wake turbulent flow of a freely rotating swept rotor blade and compares it with a straight rotor blade to shed light on the differences in their near-wake characteristics.

## 2. Methodology

### 2.1. Rotor blade characteristics

Two different three-bladed (configured similarly to horizontal axis wind turbine (HAWT)) rotor models were studied in this experiment; a straight blade and a 15° sweep angle (swept rotor) blade. Schematic illustrations of the rotor blades are shown in the isometric view in figure 1. Both blades had a taper ratio of 0.7, twist angle of −7 degree and root chords of 0.022 m. The rotor diameter was 0.22 m while each blade was approximately 0.09 m long. The mean aerodynamic chord of the blades was approximately 0.02 m. The blades were designed using NACA 4 digits aerofoil (NACA 4418), which offers a relatively higher maximum lift coefficient (1.43) for low aspect ratio rotor blades [38]. Onyx plastic (density, $\rho = 1180$ kg m$^{-3}$) was used to 3D-print the rotor hub as well as the blades.

### 2.2. Experimental set-up

The experiments were conducted in a water flume at the Environmental Fluids Laboratory in Coastal Carolina University. The flume cross-sectional dimensions are $0.7 \times 0.5$ m and a trough length of 15 m. A centrifugal pump recirculates the water through two large reservoirs located at the entrance and exit of the flume. The water height was 0.38 m in the flume to ensure the submerged model is away from both the surface and bottom boundary layer flow effects. The flow speeds tested are listed in table 1.

The measurements were performed using particle image velocimetry (PIV) [43] in the streamwise-normal plane, $0.5D$ downstream behind the model encompassing a region of $1.6D$ ($D$ = rotor diameter, 0.22 m) horizontally and $1D$ vertically for both blades' models. As the near wake is characterized by strong turbulence heterogeneity [21], the measurement region was chosen to accommodate the PIV resolution limitation enabling to resolve the smallest turbulent scales within the flow. The PIV system consists of a dual pulsed Nd:YAG laser (Quantel Inc.) operating at a wavelength of 532 nm and a 29 MP double exposure CCD camera (PowerView™) with dynamic range of 12 bits operating at 1 Hz. A 200 mm camera lens with a 20 mm extension tube was mounted on the camera yielding a field of

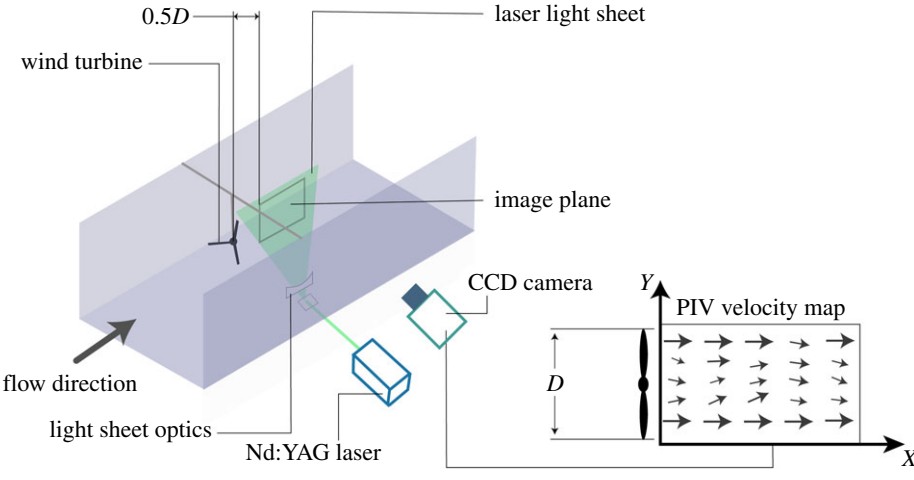

**Figure 2.** Schematic diagram of the PIV set-up within the water flume.

**Table 1.** Rotor power characteristics.

| case | $U_\infty$ (m s$^{-1}$) | Reynolds number, $Re^a$ | tip speed ratio, $\lambda^b$ | | extractable power in the freestream flow, $P_f^c$ (mW) | power output of the rotor, $P_w{}^d$ (mW) | |
|---|---|---|---|---|---|---|---|
| | | | straight | swept-back | | straight | swept-back |
| 1 | 0.12 | 23 800 | 2.53 | 1.92 | 27.14 | 9.30 | 6.83 |
| 2 | 0.15 | 29 700 | 3.28 | 2.10 | 53.01 | 18.20 | 11.82 |
| 3 | 0.18 | 35 700 | 3.78 | 2.21 | 91.61 | 31.53 | 18.00 |
| 4 | 0.21 | 41 600 | 4.34 | 2.30 | 145.47 | 50.25 | 25.04 |

[a]$Re = U_\infty L / \vartheta$, where $U_\infty$ is the freestream velocity, $L$ is a characteristic length (rotor diameter), and $\vartheta$ is the kinematic viscosity of water.

[b]$\lambda = \omega R / U_\infty$, where $\omega$ is the blade rotation per second, and $R$ is the radius of the rotor.

[c]$P_f = 1/2 \rho A U_\infty^3$, where $\rho$ density of water, $A$ is the swept area of the rotor [39].

[d]$P_w = C_p \rho A U_1^2 (U_\infty - U_2)$, where, $U_1$ is the fluid velocity passing through the rotor disc, $U_2$ is the mean velocity of the fluid downstream of the rotor. $U_1$ is estimated as: $U_1 = 1/2(U_\infty + U_2)$ [40]. $C_p$ is the power coefficient and estimated as: $C_p = 0.593 - 0.565e$, where $e = C_d/C_l$. [41,42].

view of $38 \times 28$ cm. Each experiment consisted of 500 PIV image pairs to allow for statistical convergence. The time delay between two laser pulses was set to 2360 μs and the light sheet thickness was approximately 1 mm. The streamwise ($u$) and normal ($v$) velocity components were estimated using cross-correlation analysis of the captured image pairs. A schematic diagram of the PIV set-up within the flume is presented in figure 2.

# 3. Results and discussion

## 3.1. Near-wake mean flow characteristics

The utilization of different blade configurations impacts the rotor blade performance and efficiency as they control the energy transfer from the incoming flow to the blades by changing the kinetic energy stored in the flow to rotational kinetic energy of the blade, and for an actual wind turbine it would later convert to mechanical energy on the rotor shaft. Therefore, the wake region behind these blades can provide insight as to how the flow is modulated by the blades' geometry. Here, we characterize the mean flow properties at the wake region.

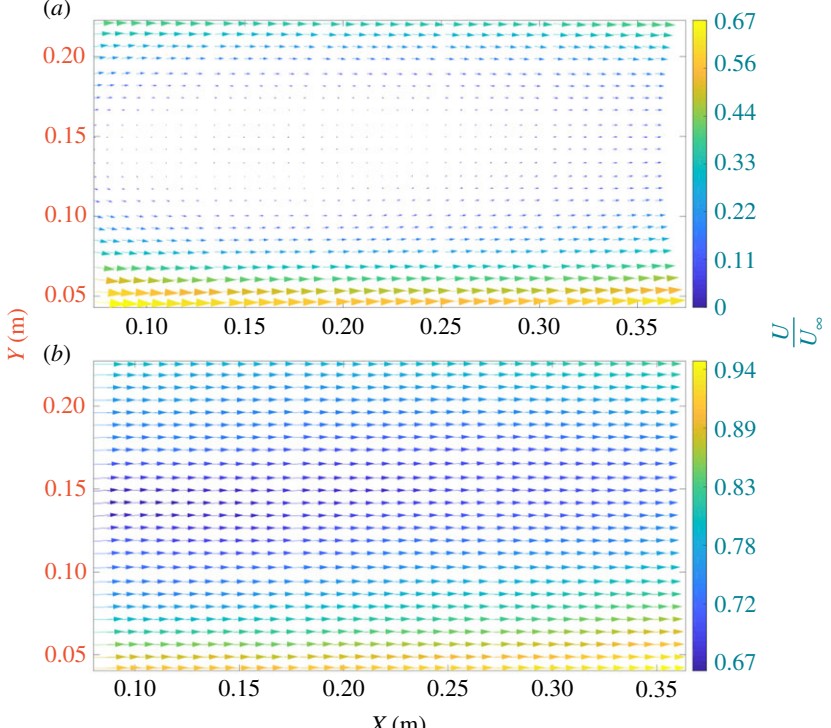

**Figure 3.** Mean velocity vectors behind (*a*) straight blade, and (*b*) swept-back blade. The *x*-axis denotes to streamwise position of vectors behind the rotor blades at 0.5*D* distance away from the blades, while *y*-axis denotes to vertical positions of vectors.

Figure 3 depicts the mean velocity maps behind the (*a*) straight and (*b*) swept-back blade operating at 0.18 m s$^{-1}$ freestream velocity (*Re* = 35 700). The *x*-axis and *y*-axis denote the streamwise and vertical positions of the vectors, respectively, at 0.5*D* distance downstream of the rotor blades. The range of the normalized velocities is different since the momentum deficit at the wake behind the swept blade is significantly smaller; hence, we had to use two different velocity magnitudes ranges. In this study, entire vertical profiles are within the vertical extent of the wake, with the mid-point of the *y*-axis being the hub location. The mean velocity field is computed from averaging 500 instantaneous wake velocity fields for each case. For both cases, it can be noticed that the magnitudes of velocities in the wakes are considerably lower than the freestream velocity, which signifies the removal of kinetic energy from the freestream by the rotor blades. The wakes of both blades are bigger than the PIV field of view; as a result, the tip vortices are not fully enveloped by the vertical axis. The slight asymmetry presented in the figure between the top and bottom regions is presumably due to the imperfections of the model as well as potential interaction with the bottom boundary layer of the flume (although the distance of the blade tip from the bottom of the flume was more than 20% of the water height). The wake of the straight blade exhibits a strong recirculation region whereas the swept-back blade wake does not. This phenomenon might be attributed to the straight blade profile which may have been subjected to a critical angle of attack at the blade root due to the high effective angle of attack [44,45]. On the other hand, the swept-back blade does not show evidence of strong flow separation, and its wake exhibits nearly constant velocity deficit at different streamwise positions. The tip speed ratios for both rotor blades are lower (table 1) than the optimal tip speed ratio for three-bladed rotors (optimal tip speed ratio for a three-bladed rotor is 6–8, [40]) at all freestream velocities; at these low tip speed ratios, the tip vortex system does not interact much with neighbouring hub vortices [46].

It can be observed in figure 3 that highest velocity magnitudes (also shown in figure 4) as well as the highest velocity gradients exist downstream of the blade tip region (green and yellow colour) that is located at the largest and smallest *y*-positions. This part of the wake contains tip vortices which can be considered, conceptually, as a cylindrical shear layer, forming an illustrative envelope, that separates the low velocity region of the wake from the freestream. The swept blade shows the thickening of this layer downstream, which has been linked to turbulent diffusion [47]. Thus, these results may suggest that the swept blade exhibits more mixing based on this increased thickness at larger streamwise positions.

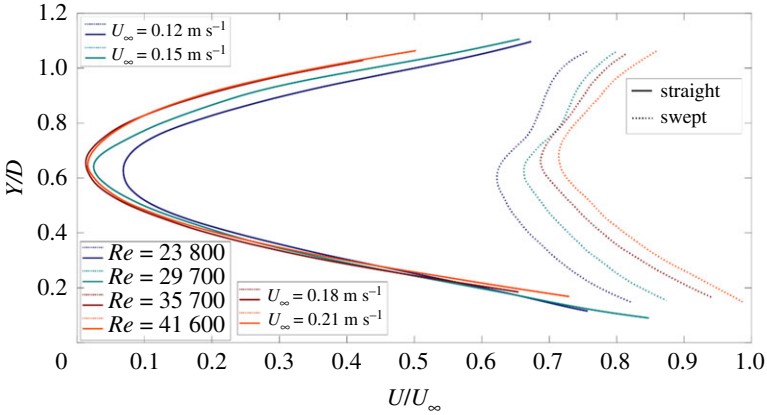

**Figure 4.** Velocity deficit behind straight blade (solid lines) and swept-back blade (dashed lines). The *x*-axis represents streamwise velocity normalized by freestream velocity and the *y*-axis represents vertical coordinates normalized by the rotor diameter (*D*).

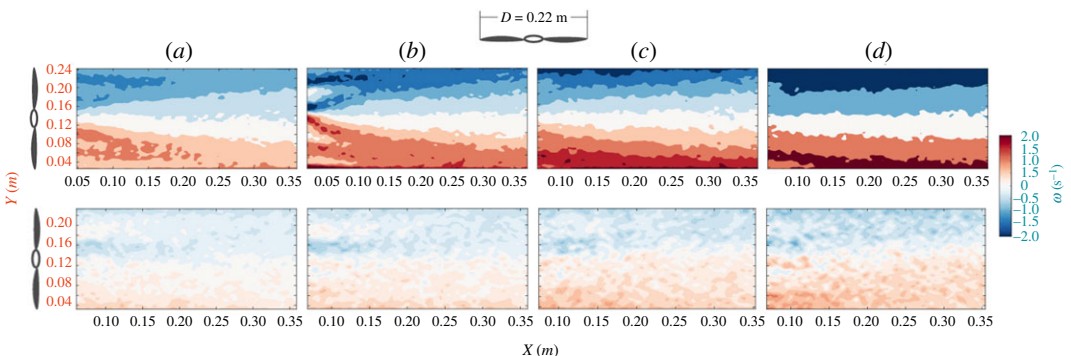

**Figure 5.** Average vorticity contour of straight (top row) and swept-back (down row) blade where each column from left represents (*a*) *Re* = 23 800, (*b*) *Re* = 29 700, (*c*) *Re* = 35 700 and (*d*) *Re* = 41 600.

For a freely rotating blade, the tip speed ratio is indicative of rotor performance and it implies how much energy is being extracted from the freestream flow [40]. In the case of similar Reynolds number, the tip speed ratio for the swept blade is significantly lower than the straight blade, which may explain the velocity deficit differences for the two rotor blades (figure 4) as the extraction of kinetic energy from the freestream will affect the magnitude of deficit. Figure 4 depicts the mean velocity profiles behind the straight (solid lines) and the swept blade (dashed lines) operating at the range of Reynolds numbers (*Re*) (listed in table 1) where the *x*-axis shows the mean streamwise velocity normalized by the freestream velocity, and the *y*-axis shows vertical coordinates normalized by the rotor diameter (*D*). This figure demonstrates a large deficit behind the hub section of the straight blade as compared with the swept one. As the near wake is directly influenced by the presence of the rotor, blade profile and hub geometry mainly affect the flow field [48]. It appears that the upstream induction zone of the straight blade causes a significant reduction in wind speed near the hub which in conjunction with blade stall, gives rise to such a deficit profile; by contrast, the deficit in the near wake of the swept blade is much smaller. These differences become critical when considering the aerodynamic loads exerted on the blade, as will be shown in the following sections. In addition, it seems that the deficit behind the swept blade is more sensitive to Reynolds number. This is somewhat contradictory to the tip speed ratios (table 1) for the swept blade, which seems to not change over the range of Reynolds number. The straight blade demonstrates similar trends as the Reynolds number increases; at 35 700 and 41 600 Reynolds number, both profiles are almost similar, which can be interpreted as its incapacity of extracting more power from the freestream with a reduction in loss factors (e.g. tip/root vortex loss).

The contour plots of the mean spanwise vorticity are represented in figure 5. The spanwise component of the vorticity, $\omega_z$ is defined as

$$\omega_z = \frac{\partial v}{\partial x} - \frac{\partial u}{\partial y},$$ (3.1)

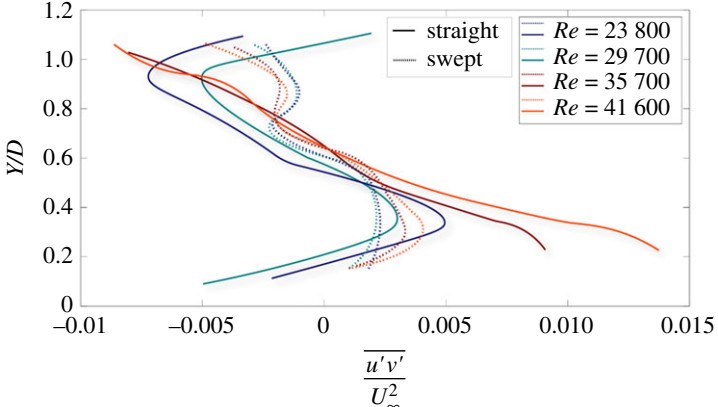

**Figure 6.** Normalized vertical profiles of normalized Reynolds shear stress at the near wake of straight blade (solid lines), and swept-back blade (dashed lines). The x-axis represents the Reynolds shear stress normalized by the freestream kinetic energy, and the y-axis represents the vertical coordinates normalized by the diameter of the rotor.

where $u$ and $v$ are the instantaneous streamwise and normal components of velocity, respectively. The vorticity is calculated using a least-squares differentiation scheme [43]. Figure 5 shows the mean spanwise vorticity contours for the four different flow speeds (table 1). The contour plots for both blades show a pattern associated with a cross-section of a so-called helical envelope characterized by strong velocity gradients, which results from the rotational motion of the blades, i.e. marked as the rotor wake [49]. The mean vorticity is associated with the magnitude of the rotational force and the pattern of its distribution into the wake. Figure 5 indicates that a higher magnitude of vorticity is produced at the tip region than the root or hub area for both types of rotors. The mean high-vorticity regions for the straight blade define the presence of a wake with two distinct shear layers, while for the swept-back blade, these are broader in space and weaker in magnitude. These discrepancies may suggest that the swept-back blades during rotation extract less rotational energy from the flow, which is also consistent with the lower r.p.m. and tip speed ratio for the swept-back blade observed during the experiments.

## 3.2. Wake turbulence characteristics

Our main interest is to explore how a swept-back blade configuration for a freely rotating rotor may impact the wake flow field and lead to changes in its aerodynamic performance. This performance depends on the state of the near-wake flow region. The wake of rotating blades, given the high Reynolds numbers, is inherently turbulent. Therefore, we characterize how turbulence changed in the wake region for these blades as compared with classical blades. The turbulence stresses and turbulence kinetic energy components are presented herein.

Figure 6 depicts vertical profiles of normalized Reynolds shear stress (turbulent momentum flux) in the near wake of both rotor blades operating at various freestream Reynolds numbers. The x-axis represents the Reynolds shear stress ($\overline{u'v'}$, where $u'$ is streamwise velocity fluctuations and $v'$ is normal velocity fluctuations) normalized by the freestream kinetic energy ($U_\infty^2$), and the y-axis represents the vertical coordinates normalized by the diameter of the rotor blade. Irrespective of speeds, the straight blade shows higher shear stress magnitude compared with the swept blade. Among all cases, the maximum momentum flux is observed for the straight blade case operating at the maximum Reynolds number (41 600). Further, irrespective of blades, the Reynolds stresses reach their peak around the tip areas. The trend of the Reynolds stress profile in the near-wake region is similar to turbulent wakes, where the profile varies from a maximum at the entrainment region towards a minimum at the core, which also follows the mean spanwise vorticity profile variation [50]. The rotor motion enhances the Reynolds stresses by the entrainment of turbulence momentum from the tip and redistributing it towards the hub [48]; the stresses are highest around the tip area where the shear is the strongest. The negative and positive values of the stresses represent the redistribution of the turbulent momentum flux radially towards the hub. The trend of the Reynolds stress profiles matches with the turbulent momentum flux behaviour in rotor wakes reported in Barlas *et al.* [51] and Wu & Porté-Agel [52]. However, the most noticeable result is the narrow range of Reynolds stress at

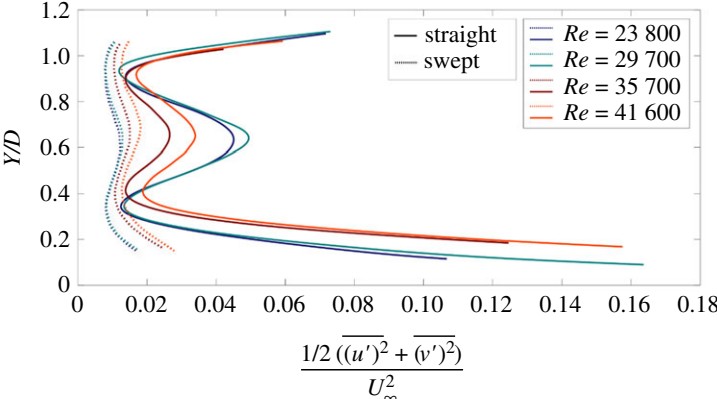

**Figure 7.** Normalized vertical profiles of normalized turbulent kinetic energy at the near wake of straight blade (solid lines), and swept-back blade (dashed lines). The *x*-axis represents the turbulent kinetic energy normalized by the freestream kinetic energy, and the *y*-axis represents the vertical coordinates normalized by the diameter of the rotor.

the wake of the swept blade for all speeds tested as compared with the straight blade. These differences may suggest that the turbulent activity in the near-wake region of the swept blade is smaller; thus, reducing the level of available turbulent energy, as will be shown in the next section, which may impact the overall performance. We also note from figure 5 that the Reynolds stresses exhibit dependency on Reynolds number in the tip regions, which is not surprising given that this is the entrainment region which is characterized by strong shear.

Vertical profiles of normalized turbulent kinetic energy (TKE) are compared in figure 7 for the straight and swept blades. The TKE is based on two fluctuating velocity components (streamwise and normal) and estimated as $1/2(\overline{u'^2} + \overline{v'^2})$. The *x*-axis represents the turbulent kinetic energy normalized by the freestream kinetic energy, and the *y*-axis represents the vertical coordinates normalized by the diameter of the rotor blade. Irrespective of Reynolds number, TKE in the wake of the straight blade is significantly higher than the swept blade. The profiles of the straight blade exhibit that the most pronounced turbulent activity occurs around the tip and steeply reduces toward the hub area. This result matches the observation from the Reynolds stress profiles that turbulence levels and turbulent momentum flux are higher around the blade tip. In terms of TKE magnitude, two minima appear at $Y/D \sim 0.9$ and $Y/D \sim 0.3$ with a local maximum observed at $Y/D \sim 0.6$ where the hub is located. This trend appears for both blade geometries, although it is less pronounced for the swept blades. The occurrence of a local maximum represents the hub vortex with its core located at the centre of the hub. The two minima observed mark the onset of entrainment region, which is also consistent with the Reynolds stresses distribution. Thus, the turbulent kinetic energy is similar to the trend depicted by the Reynolds stresses. The suppression of TKE for the swept blade suggests that the turbulent activity in the near wake of the swept blade is significantly reduced compared with the straight blade despite the flow velocities in the wake of the swept blade being higher.

In figure 8, vertical profiles of turbulent kinetic energy production (TKEP) are shown for both blades operating at different Reynolds numbers. The TKEP is calculated as $\overline{u_i' u_j'} S_{ij}$ which consists, in our case, of four components (out of nine) calculated from the PIV data: $\overline{u'u'}S_{11}$, $\overline{v'v'}S_{22}$ and $2\overline{u'v'}S_{12}$ (i.e. $S_{12} = S_{21}$), where mean shear strain, $S_{ij} = 1/2(\partial U_i/\partial x_j + \partial U_j/\partial x_i)$; $U$ is the mean velocity field, $i$ and $j$ are indices corresponding to 1 and 2 or *x*- and *y*-directions in a Cartesian coordinate system. We notice that the straight blade demonstrates two maxima occurring around the tip area and reduces toward the hub for each Reynolds number case. This observation matches the behaviour of the Reynolds stress and the TKE profiles as well: both properties display pronounced turbulent activity around the tip and steep reduction toward the hub. For all Reynolds numbers, the profiles of the swept blade display significantly lower production values compared with the straight blade, suggesting suppressed turbulence generation in the near wake of the swept blade. Overall, the production profiles reinforce that the turbulent activity in the near wake of the swept blade is suppressed compared with the straight blade and for each Reynolds number case, the maximum activity occurs around the entrainment region (tip) for both blades.

Comparisons of turbulence profiles demonstrate that the swept blade exerts less turbulent stress into the wake flow compared with the straight blades at all Reynolds numbers. Commonly, higher turbulence

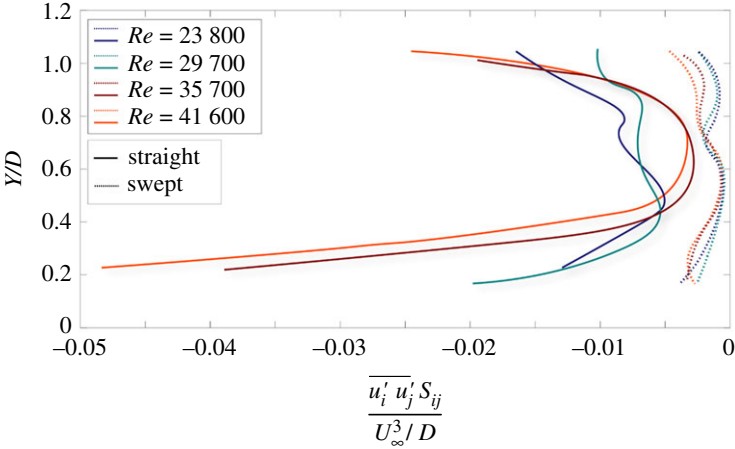

**Figure 8.** Normalized vertical profiles of the turbulent kinetic energy production term in the near wake of straight blade (solid lines) and swept blade (dashed lines). The x-axis shows the production term normalized with the ratio between the cube of freestream speed to the rotor diameter, and the y-axis shows the vertical coordinates normalized by the diameter of the rotor.

increases load fluctuations on blades [4], which has negative effects on the performance of the blades in converting energy from the incoming flow. In some cases, inflow turbulence has been reported to shorten the wake recovery process by aiding in breaking down the entrainment zone [53]. Swept rotor blades have been shown to reduce load fluctuations [54]. However, the reduction of turbulence activity in the wake region sometimes may have an inverse effect as it can inhibit faster wake recovery and allow a longer wake signature in time and space [55].

## 3.3. Aerodynamic loads

### 3.3.1. Sectional drag

Streamwise velocity vertical profiles were sampled at various streamwise positions downstream of the rotor within each instantaneous velocity field map. These instantaneous profiles for each PIV vector map were spatially averaged over all streamwise positions yielding one vertical profile, referred to as an ensemble momentum deficit, similar to the procedure described in Ben-Gida *et al.* [56]. This ensemble momentum deficit is used to estimate an 'instantaneous' drag per unit span ($D'$) for each PIV velocity map [57] following equation:

$$D' = \rho \int_0^h u(U_\infty - u)\, \mathrm{d}y. \qquad (3.2)$$

The drag was calculated using equation (3.2) for every instantaneous velocity field of the rotor blade wake, where $u$ is the spatially averaged 'instantaneous' streamwise velocity, $U_\infty$ is the freestream velocity, $r$ is the water density, $h$ is the vertical length of the computed velocity field in the wake, and $\mathrm{d}y$ corresponds to the vertical spacing between two data points. The drag (equation (3.2)) was calculated for every instantaneous velocity field of the rotor blade wake. The process involves a spatial window averaging that smooths out some variations within each vector map. Subsequently, all 500 'instantaneous' sectional drag values were averaged to estimate a single profile drag for every $Re$ for both the rotor blades.

### 3.3.2. Sectional lift

In order to estimate the lift, we will use the Kutta–Joukowski (K–J) theorem. The theorem relates the lift generated by an aerofoil to the speed of fluid flow ($U_\infty$), the density of the fluid ($\rho$) and the circulation ($\Gamma$) around the aerofoil. The theorem applies to the two-dimensional flow around an aerofoil and lift per unit span, $L'$ is given by

$$L' = -\rho U_\infty \Gamma. \qquad (3.3)$$

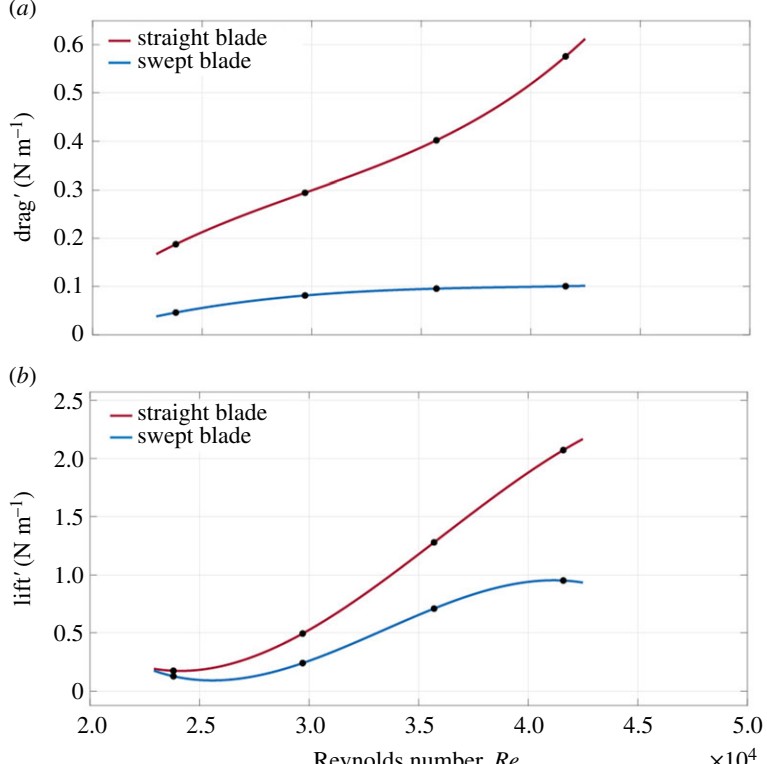

**Figure 9.** Sectional drag (a) and sectional lift (b) for straight and swept blade over Reynolds number (Re).

The circulation ($\Gamma$) is a measure of rotation for a finite area ($A$) of the fluid domain, and can be expressed in terms of the vorticity field [58]. The circulation is estimated from the surface integral of vorticity using Stokes' theorem:

$$\Gamma = \iint_A \omega_z \,.\, dA, \tag{3.4}$$

where $\omega_z$ is the mean spanwise vorticity and $A$ is the area of the near wake. Estimation of circulation (equation (3.4)) using spanwise vorticity (equation (3.1)) in the wake of an aerofoil would quantify the circulation around the aerofoil [59,60]. For each case, we calculate sectional lifts (equation (3.3)) for all 500 instantaneous PIV maps, which are then averaged to estimate a single-sectional lift for each blade geometry and $Re$.

### 3.3.3. Lift and drag estimates

The torque produced by the blades in a rotor results from the lift generated by each blade. For each blade, lift force results from the pressure difference between the upper and lower surface of the blade. Tip vortices result from spanwise lift variations of each blade and the strength of the tip vortices increases linearly with the increase of lift force generation; however, the radius of the tip vortices is not influenced by the lift force [61]. An important characteristic of the swept wing is that the effective component of the freestream velocity ($U_\infty\cos\varphi$, where $\varphi$ is the sweep angle) is significantly reduced due to the sweep angle, which results in lower lift generation; hence, while designing a swept wing, it was recommended that either wing surface area or local angle of attack should be higher than the conventional wing [62] or alternatively increase the sweeping angle of the blade [28]. In this study, the swept blade area as well as the local angles of attack were kept similar to the straight wing and the sweeping angle was relatively low, which might explain the lower lift generation of the swept wing as compared with the straight one, shown in figure 9b.

Theoretically, for a lift-based rotor, as the drag is reduced by a factor of 2, the optimal power coefficient increases by a factor of 4 [63]. Hence, rotor blades are designed with aerofoils that have high lift-to-drag ratios. We can observe from figure 9a that the straight-bladed rotor generates more

**Table 2.** Aerodynamic performances for straight and swept-back blades.

| case | Reynolds number, Re | average sectional drag (N m$^{-1}$) | | average sectional lift (N m$^{-1}$) | | lift-to-drag ratio (lift'/drag') | |
|---|---|---|---|---|---|---|---|
| | | straight | swept-back | straight | swept-back | Straight | swept-back |
| 1 | 23 800 | 0.18 | 0.05 | 0.18 | 0.13 | 1.00 | 2.60 |
| 2 | 29 700 | 0.29 | 0.08 | 0.50 | 0.24 | 1.72 | 3.00 |
| 3 | 35 700 | 0.40 | 0.09 | 1.28 | 0.71 | 3.20 | 7.88 |
| 4 | 41 600 | 0.58 | 0.10 | 2.07 | 0.95 | 3.57 | 9.50 |

drag as compared with the swept-bladed rotor. At $Re = 23\,800$, swept-bladed rotor produces 0.05 N m$^{-1}$ drag force, whereas the straight blade generates 0.18 N m$^{-1}$ drag force. As the Reynolds number increases, the straight blade exhibits a steep rise in the drag curve, whereas the swept blade demonstrates only a slight increase in drag. This result is significant, as profile drag is associated with momentum loss downstream the wake, which affects the energy conversion level; thus reducing the amount of power that can be extracted. Based on table 1, the power output of the straight blade surpasses the swept one. For the same Reynolds number, there is a rapid increase in tip speed ratio, which reflects the sectional lift curves as shown in figure 9b. However, while the straight blade generates more power, the lift-to-drag ratio of the swept blade is larger than the straight blade (table 2). Higher lift'/drag' ratio may be associated with the blade's performance to accumulate operational torque while maintaining blade bending moments and associated energy losses at low values [64]. From figure 9b, we observe that lift generated by the straight blade mostly follows a linear trend, whereas the swept blade mostly demonstrates a nonlinear trend. Also, at low Reynolds number, the straight blade generates almost the same amount of lift as the swept one, but the difference in drag is significant.

## 4. Conclusion

The flow characteristics in the near wake of a freely rotating swept blade model and its aerodynamic loads have been examined. The experiments included using a similar rotor model with straight blades as a reference for comparison purposes. We have shown that the near-wake flow characteristics of the swept rotor blade are substantially different compared with the wake developed behind the straight rotor blade. The momentum deficit in this region for the swept blade is small as are the turbulence kinetic energy and stresses relative to the straight rotor blade. The straight blade generates more lift, yet at the expense of more drag, as shown in table 2 and figure 9. The *lift'/drag'* ratio is larger for the swept blade model, which provides some indication of its aerodynamic efficiency, and it increases dramatically as $Re$ increases. The relatively lower lift generated by the swept blade may be associated with its swept angle that presumably can be further optimized to enhance lift [28]. These outcomes can be helpful when wake losses are considered in the design process of rotating blades (e.g. wind farm, multirotor vehicles, tiltrotor aircraft). Specifically, in cases where structures/rotating blade operates in the wake of an upstream rotor blade; they suffer from degradation in their overall performance. This scenario occurs due to the reduction of available energy for the downstream system(s) caused by the shedding effects of an upstream rotating blade. Our results suggest that the utilization of swept blades, inspired by nature, may reduce wake losses because they cause a minimal downstream disturbance at the expense of producing lower lift values.

Data accessibility. All data analysed during this study can be accessed at the Dryad Digital Repository: https://doi.org/10.5061/dryad.xksn02vdb [65].

Authors' contributions. A.S.N, K.K. and A.K.D.: experiments, formal analysis., A.S.N, A.D., E.E.H. and R.G.: methodology, writing - review and editing.

Competing interests. We declare we have no competing interests.

Funding. We received no funding for this study.

Acknowledgements. A.S.N. and A.K.D. wish to thank CMSS department at Coastal Carolina University for supporting their research.

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
