## [Peer Review File · Royal Society Open Science]

Review History

RSOS-201597.R0 (Original submission)

Review form: Reviewer 1

Is the manuscript scientifically sound in its present form?

Yes

Are the interpretations and conclusions justified by the results?

Yes

Is the language acceptable?

Yes

Do you have any ethical concerns with this paper?

No

Have you any concerns about statistical analyses in this paper?

No

Recommendation?

Major revision is needed (please make suggestions in comments)

Comments to the Author(s)

Wake characteristics behind a horizontal axis wind turbine were focused in this study, and two types of blades were compared. The experiments were conducted in a water flume with PIV technique. For the two wind turbine models, the near wake mean flow characteristics, the wake turbulence properties, and the aerodynamic loads were analyzed. However, some shortcomings must be improved before possible publication.

Major comments:

1. Wind tunnel experiments were conducted to investigate wind turbine performance in most cases. In this study, the experiments were conducted in a water flume, so it is very necessary to validate its accuracy.
2. In section 4.1, the near wake mean flow characteristics were discussed, but the tested region is limited. In the vertical direction, the wake characteristics in the outside region of the blade tip are also important. In the horizontal direction, a longer distance is needed to better understand the wake recovery which is closely related to the issue discussed in this paper.

Minor comments:

3. In section 3.1, the dimensions discussed in the content should be marked in Fig. 1 to help readers better understand what they represent.
4. Some pictures of the real wind turbine models in the water flume should be given.
5. Why the water height of 0.38 m was selected? Is it enough for the wind turbine models?
6. Why a distance of 0.5D was set between the turbine hub and the measurement plane?

Review form: Reviewer 2

Is the manuscript scientifically sound in its present form?

No

Are the interpretations and conclusions justified by the results?

No

Is the language acceptable?

Yes

Do you have any ethical concerns with this paper?

No

Have you any concerns about statistical analyses in this paper?

No

Recommendation?

Reject

Comments to the Author(s)

See the attached file (Appendix A).

Decision letter (RSOS-201597.R0)

This year has been very difficult for everyone, and we want to take the opportunity to thank you for your continued support in 2020.

The Royal Society Open Science editorial office will be closed from the evening of Friday 18 December 2020 until Monday 4 January 2021. We will not be responding during this time. If you have received a deadline within this time period, please contact us as soon as possible to allow us to extend the deadline. If you receive any automated messages during this time asking you to meet a deadline, we offer apologies and invite you to respond after the festive period or during normal working hours.

With our best for a peaceful festive period and New Year, and we look forward to working with you in 2021.

Dear Mr Nafi

The Editors assigned to your paper RSOS-201597 "Wake characteristics behind a horizontal axis wind turbine with swept-back blades" have made a decision based on their reading of the paper and any comments received from reviewers.

Regrettably, in view of the reports received, the manuscript has been rejected in its current form. However, a new manuscript may be submitted which takes into consideration these comments.

We invite you to respond to the comments supplied below and prepare a resubmission of your manuscript. Below the referees' and Editors' comments (where applicable) we provide additional requirements. We provide guidance below to help you prepare your revision.

Please note that resubmitting your manuscript does not guarantee eventual acceptance, and we do not generally allow multiple rounds of revision and resubmission, so we urge you to make every effort to fully address all of the comments at this stage. If deemed necessary by the Editors, your manuscript will be sent back to one or more of the original reviewers for assessment. If the original reviewers are not available, we may invite new reviewers.

Please resubmit your revised manuscript and required files (see below) no later than 14-Jun-2021. Note: the ScholarOne system will 'lock' if resubmission is attempted on or after this deadline. If you do not think you will be able to meet this deadline, please contact the editorial office immediately.

Please note article processing charges apply to papers accepted for publication in Royal Society Open Science (<https://royalsocietypublishing.org/rsos/charges>). Charges will also apply to papers transferred to the journal from other Royal Society Publishing journals, as well as papers submitted as part of our collaboration with the Royal Society of Chemistry (<https://royalsocietypublishing.org/rsos/chemistry>). Fee waivers are available but must be requested when you submit your manuscript (<https://royalsocietypublishing.org/rsos/waivers>).

Thank you for submitting your manuscript to Royal Society Open Science and we look forward to receiving your resubmission. If you have any questions at all, please do not hesitate to get in touch.

Kind regards,
Andrew Dunn

Royal Society Open Science Editorial Office
Royal Society Open Science
openscience@royalsociety.org

on behalf of Prof R. Kerry Rowe (Subject Editor)
openscience@royalsociety.org

Associate Editor Comments to Author:

Thank you for this paper, which has now been reviewed by two referees. On careful review of the commentary received, it is clear that the manuscript is not yet at a stage that publication would be appropriate. However, both reviewers offer a number of ways you may strengthen the research. That a number of the suggestions will require further experimentation leads us to recommend a rejection at this stage with an option to resubmit when you are ready to do so, should give you sufficient time to make the changes needed. Good luck and we'll look forward to receiving the resubmission in due course.

Reviewer comments to Author:

Reviewer: 1

Comments to the Author(s)

Wake characteristics behind a horizontal axis wind turbine were focused in this study, and two types of blades were compared. The experiments were conducted in a water flume with PIV technique. For the two wind turbine models, the near wake mean flow characteristics, the wake turbulence properties, and the aerodynamic loads were analyzed. However, some shortcomings must be improved before possible publication.

Major comments:

1. Wind tunnel experiments were conducted to investigate wind turbine performance in most cases. In this study, the experiments were conducted in a water flume, so it is very necessary to validate its accuracy.
2. In section 4.1, the near wake mean flow characteristics were discussed, but the tested region is limited. In the vertical direction, the wake characteristics in the outside region of the blade tip are also important. In the horizontal direction, a longer distance is needed to better understand the wake recovery which is closely related to the issue discussed in this paper.

Minor comments:

3. In section 3.1, the dimensions discussed in the content should be marked in Fig. 1 to help readers better understand what they represent.
4. Some pictures of the real wind turbine models in the water flume should be given.
5. Why the water height of 0.38 m was selected? Is it enough for the wind turbine models?
6. Why a distance of $0.5D$ was set between the turbine hub and the measurement plane?

Reviewer: 2

Comments to the Author(s)

See the attached file

===PREPARING YOUR MANUSCRIPT===

a 'clean' version of the new manuscript that incorporates the changes made, but does not highlight them. This version will be used for typesetting if your manuscript is accepted. Please ensure that any equations included in the paper are editable text and not embedded images.

===PREPARING YOUR REVISION IN SCHOLARONE===

- If you are requesting a discretionary waiver for the article processing charge, the waiver form must be included at this step.
- If you are providing image files for potential cover images, please upload these at this step, and inform the editorial office you have done so. You must hold the copyright to any image provided.
- A copy of your point-by-point response to referees and Editors. This will expedite the preparation of your proof.

- Ensure that your data access statement meets the requirements at <https://royalsociety.org/journals/authors/author-guidelines/#data>. You should ensure that you cite the dataset in your reference list. If you have deposited data etc in the Dryad repository, please include both the 'For publication' link and 'For review' link at this stage.
- If you are requesting an article processing charge waiver, you must select the relevant waiver option (if requesting a discretionary waiver, the form should have been uploaded at Step 3 'File upload' above).
- If you have uploaded ESM files, please ensure you follow the guidance at <https://royalsociety.org/journals/authors/author-guidelines/#supplementary-material> to include a suitable title and informative caption. An example of appropriate titling and captioning may be found at https://figshare.com/articles/Table_S2_from_Is_there_a_trade-off_between_peak_performance_and_performance_breadth_across_temperatures_for_aerobic_scope_in_teleost_fishes_/3843624.

Author's Response to Decision Letter for (RSOS-201597.R0)

See Appendix B.

RSOS-210779.R0

Review form: Reviewer 1

Is the manuscript scientifically sound in its present form?

Yes

Are the interpretations and conclusions justified by the results?

Yes

Is the language acceptable?

Yes

Do you have any ethical concerns with this paper?

No

Have you any concerns about statistical analyses in this paper?

No

Recommendation?

Accept as is

Comments to the Author(s)

The authors have responded my questions.

Review form: Reviewer 2

Is the manuscript scientifically sound in its present form?

Yes

Are the interpretations and conclusions justified by the results?

Yes

Is the language acceptable?

Yes

Do you have any ethical concerns with this paper?

No

Have you any concerns about statistical analyses in this paper?

No

Recommendation?

Accept as is

Comments to the Author(s)

Accept as is

Decision letter (RSOS-210779.R0)

Dear Mr Nafi,

I am pleased to inform you that your manuscript entitled "Wake characteristics of a freely rotating bio-inspired swept rotor blade" is now accepted for publication in Royal Society Open Science.

Please ensure that you send to the editorial office an editable version of your accepted manuscript, and individual files for each figure and table included in your manuscript. You can send these in a zip folder if more convenient. Failure to provide these files may delay the

processing of your proof. You may disregard this request if you have already provided these files to the editorial office.

on behalf of R. Kerry Rowe (Subject Editor)
openscience@royalsociety.org

Associate Editor Comments to Author:
Comments to the Author:
Congratulations! The reviewers recommend acceptance.

Reviewer comments to Author:
Reviewer: 1
Comments to the Author(s)
The authors have responded my questions.

Reviewer: 2
Comments to the Author(s)
accept as is

Appendix A

Review of the paper “Wake characteristics behind a horizontal axis wind turbine with swept-back blades”

The paper reports experimental results regarding the comparison of a wind-turbine model with a traditional tapered blade and one with an innovative design where a sweep is introduced. While the topic is quite interesting with a good idea, the experimental setup and data analysis is unfortunately not well performed since a comparison of the two blades is not really possible as done here. This is elaborated in my comments here below. The overall manuscript is written with some basic understanding of actuator disk theory without considering alternative theories such as general momentum theory and blade-element momentum theory. Therefore I cannot recommend the publication of the manuscript in its present form and I suggest a careful re-planning of the blade design and experiments.

1. How were both blades designed? Every blade necessitates the definition of the airfoil, the chord and the twist angle of each section. These informations are not provided here. This paper would have been much more interesting if, given the same properties (airfoil, chord and twist angle), the swept blade had a different sweep angle only with respect to the standard (straight) blade (according to BEM theory, the two blades should be the same). I suspect this is not the case here, although this was not specified by the authors.
2. It is very interesting that the ratio between the power estimated for the straight rotor P_w and the power of the free stream, P_f , is very close to the Betz limit (at least for the straight blade). For all the rotor model I have investigated, I have never exceeded a power coefficient of 0.4. Utility wind turbines oscillate between 0.43 - 0.5 depending on the accounted losses. Therefore I cannot trust the estimate they made with P_w to be a reliable approximation of the actual power extracted by the turbine. Why was the actual power output not measured?
3. In a generic wind turbine the thrust is very important as well. Here it was not measured but the authors estimated that with a 2-d estimate of the drag (not even axisymmetric, so quite wrongly balanced). One of my main objections is how we can compare these two different blades. One could compare two turbines with the same thrust coefficient or at least with the same power coefficient or tip-speed ratio. Right now the

two rotors share only the same free-stream velocity and the comparison is performed on things that are unrelated to the operating condition (such as the velocity deficit). For instance, consider the straight blade not rotating: the wake will be much weaker and according the analysis of the authors this case should be better in terms of wake deficit.

4. The estimate of U_2 at page 5 relies too much in actuator disk theory. Real rotors create a turbulent wake and the wake starts to recover back to the free-stream velocity so that the wake velocity is x dependent.
5. Figure 3 should be normalised by the free-stream velocity as well (as done later on in figure 4). However, as the authors noted, the deficit is different and I suspect this is due to the different thrust coefficients the turbines have. I think that a better comparison could have been to have the same thrust coefficient and check which one had the highest power coefficient. The latter requires an assessment of the angular velocity of the rotor and the torque and necessitates of another measurement technique (just monitoring the generator) or of stereo-PIV (if the most complicated way possible is chosen).
6. Figures 4, 5, 7 and 8 are normalised with the free-stream velocity. Now again I think that since the two rotors have different thrust coefficients, a fair comparison is not done by using the free-stream velocity as a reference scale. I think instead that the velocity field should be treated in terms of velocity deficit $U_\infty - U(x, y)$ and normalised by the maximum velocity deficit measured somewhere (in agreement with a Galilean invariance principle). This deficit velocity scale should be used to scale the turbulence statistics as well.
7. Page 7. The statement about the tip-speed ratio is new to me. I am aware of general momentum theory and Glauert optimum result, but if we consider freely-rotating rotors, they rotate the fastest and do not produce anything.
8. Turbulence is not always negative. For wind farms, for instance, it provides a faster wake recovery and a shorter wake, enhancing the farm production. Less turbulence means longer wakes and lower power production, as observed for stable stratification conditions over wind farms.
9. How is the circulation Γ estimated? This could be done by the instantaneous PIV velocity field but most likely it will just characterise the

tip-vortex strength that is not necessarily the circulation at the blade (that by the way might not even be constant).

Appendix B

April 29, 2021

Editor, Prof. R. Kerry Rowe
Royal Society Open Science

Dear Prof. Rowe,

Enclosed please find the revised manuscript entitled “Wake characteristics of a freely rotating bio-inspired swept rotor blade” (we have revised the title to better reflect the manuscript content).

We would like to thank the editor and the reviewers for providing constructive comments that helped us improve the paper. We have performed a through revision of the analysis and the text. We have revised the content associated with wind turbines and focused our analysis and outcomes to rotating blades, which plays a critical component in converting energy in wind turbines as well as in other applications. Hence, we emphasized the choice of using bio-inspired blade configuration in improving some of the aerodynamics features associated with rotating blades.

We have modified the manuscript in response to the reviewers’ comments. In our response, we have addressed each individual comment made by the reviewers and modified the manuscript accordingly. The responds appear in the rebuttal letters. In order to follow the reviewers’ comments adequately, the rebuttal letters address their questions as appear in their original review (black font) followed by our respond (blue font).

In addition, we enclose a marked-up copy that highlight the changes made from the revised submission (marked in red font). We hope you find the revised manuscript acceptable for publication in Royal Society Open Science.

Sincerely,

Gurka Roi

Roi Gurka
Department of Physics and Engineering Science
Coastal Carolina University
290 Allied Dr., Conway, SC, 29528, USA

Response to Reviewer 1

Wake characteristics behind a horizontal axis wind turbine were focused in this study, and two types of blades were compared. The experiments were conducted in a water flume with PIV technique. For the two wind turbine models, the near wake mean flow characteristics, the wake turbulence properties, and the aerodynamic loads were analyzed. However, some shortcomings must be improved before possible publication.

Major comments:

1. Wind tunnel experiments were conducted to investigate wind turbine performance in most cases. In this study, the experiments were conducted in a water flume, so it is very necessary to validate its accuracy.

Answer: Thanks for the comment. Our current facilities allow us to test rotor blades in a water flume rather than a wind tunnel. In a similar manner, there are many studies where aircraft wing design, wind turbine design, etc. were tested in water flumes instead of wind tunnel as their aerodynamic characteristics are essentially the same in similar Reynolds number ranges (i.e., Okulov et al., 2012; Naumov et al., 2014). The non-dimensional characteristic in this study is the Reynolds number based on the chord length, and we have described all our findings in terms of Reynolds number. The Reynolds number in our experiment ranged between 23,800 and 41,600. Since we did not attempt to match a full-scale wind turbine in our current setup rather focus on the aero/hydrodynamic characteristics of bio-inspired rotating blades, the Reynolds number used here is significantly smaller compared to a full scaled wind turbine. We focus on the fluid-blade interactions which for this Reynolds range can be considered turbulent. Furthermore, the main goal here was to investigate how different a bio-inspired blade performs compared to a classical one, as we demonstrate in the results section.

References:

- Okulov, V.L., Naumov, I.N., Kabardin, I., Mikkelsen, R. and Sørensen, J.N. (2012). Experimental investigation of the wake behind a model of wind turbine in a water flume. *Journal of Physics: Conference Series*, 555, 012080.
 - Naumov, I.V., Mikkelsen, R.F., Okulov, V.L. and Sørensen, J.N. (2014). PIV and LDA measurements of the wake behind a wind turbine model. *Journal of Physics: Conference Series*, 524, 012168.
2. In section 4.1, the near wake mean flow characteristics were discussed, but the tested region is limited. In the vertical direction, the wake characteristics in the outside region of the blade tip are also important. In the horizontal direction, a longer distance is needed to better understand the wake recovery which is closely related to the issue discussed in this paper.

Answer: Thanks for the comment. We agree with the reviewer that in order to study wake recovery, we need a longer distance in the horizontal direction (i.e.: far-wake). In this study, we focus on the near wake characteristics of bio-inspired rotating blades (a one to five rotor diameter distance behind the rotor blade (McKay et al., 2012)), and how turbulence is manifested by using a blade geometry inspired by swift wings, which have been shown to be very efficient flyers over long distances and periods of time (Henningsson and Hedenström, 2011; Hedenström et al., 2016). Measuring a much larger wake region would have decreased the resolution required to resolve the turbulence scales in the near wake. Whilst this is a propelled wake and one cannot assume the turbulence will dissipate fast, it is plausible to assume that the strongest interactions occur within the near wake region (Zhang et al., 2012). We have revised the text as follows: “The measurements were0.5D downstream behind the rotor blade encompassing a region of 1.6D (D = rotor diameter, 0.22m) horizontally and 1D vertically for both the rotor blades. As the near wake is characterized by strong turbulence heterogeneity (Zhang et al., 2012), the measurement region was chosen to sufficiently resolve the turbulent fine scales”.

In the vertical direction, tip vortices are not fully enveloped by the measurement region in our experiments. As we described above, we focus on the near-wake region and its primary features (i.e., momentum deficit, Reynolds stress and turbulent kinetic energy as well as aerodynamic loads). It is noteworthy that tip vortices in the near wake are not dominant (Micallef et al., 2014). If we carefully examine the mean vorticity field obtained from our data as shown here in Figure 1, the highest magnitude mean vorticity is near the largest and smallest Y positions which could suggest the presence of tip vortices. As a reference, we add Figure 2 (Tian et al., 2014), which shows instantaneous vorticity distribution behind a wind turbine. Tian et al. (2014) also captured half of the bottom tip vortices, which was sufficient for providing qualitative information regarding the wake evolution behind rotating blades such as breakdown of tip vortices as well as wake expansion during different wind conditions.

Figure 1. Average vorticity contour of straight (top row) and swept-back (down row) blade where each column from left represents (1) $Re = 23800$, (2) $Re = 29700$, (3) $Re = 35700$ and (4) $Re = 41600$

Figure 2 :Shedding of unsteady vortex structures from a wind turbine (Tian et al., 2014). Retrieved from <https://www.aere.iastate.edu/~huhui>

References:

- McKay, P., Carriveau, R., Ting, D. and Newson, T. (2012). Turbine wake dynamics. *Advances in Wind Power*, 65.
- Henningsson, P. and Hedenström, A. (2011). Aerodynamics of gliding flight in common swifts. *Journal of Experimental Biology*, 214, 382–393.
- Hedenström, A., Norevik, G., Warfvinge, K., Andersson, A., Bäckman, J., and Åkesson, S. (2016). Annual 10-month aerial life phase in the common swift *Apus apus*. *Current Biology*, 26(22), 3066-3070.
- Zhang, W., Markfort, C.D. and Porté-Agel, F. (2012). Near-wake flow structure downwind of a wind turbine in a turbulent boundary layer. *Exp Fluids* 52, 1219–1235.
- Micallef, D., Akay, B., Ferreira, C., S., Sant, T. and Bussel, G. (2014). The origins of a wind turbine tip vortex. *Journal of Physics: Conference Series*, 555, 012074.
- Tian, W., Ozbay, A. and Hu, H. (2014). Effects of incoming surface wind conditions on the wake characteristics and dynamic wind loads acting on a wind turbine model. *Physics of Fluids*, 26, 125108.

Minor comments:

3. In section 3.1, the dimensions discussed in the content should be marked in Fig. 1 to help readers better understand what they represent.

Answer: Thanks for the comment. We have attached the dimensions in the CAD figure shown here and added to the revised document, as figure 1.

Figure 3: CAD model of rotor blade A. straight blade B. swept blade

4. Some pictures of the real wind turbine models in the water flume should be given.
 Answer: Pictures of the rotating blades in the flume are included here and added to the revised manuscript as Appendix A:

Figure 4: Freely rotating rotors in the flume during experiment, a. straight bladed rotor b. swept bladed rotor

5. Why the water height of 0.38 m was selected? Is it enough for the wind turbine models?

Answer: Thanks for the comment. The blade models (rotor diameter, $D = 0.22\text{m}$) were completely submerged in the water (0.38m height). We choose the water height to be

0.38m as this is the maximum height in the flume where we can maintain smooth surface interface. To capture the near wake flow characteristics, this water height can be considered sufficient as the wake expansion is not strong in the near wake (Micallef et al., 2014).

References:

- Micallef, D., Akay, B., Ferreira, C., S., Sant, T. and Bussel, G. (2014). The origins of a wind turbine tip vortex. *Journal of Physics: Conference Series*, 555, 012074.

6. Why a distance of $0.5D$ was set between the turbine hub and the measurement plane?

Answer: Thanks for the comment. In this study, we analyze the near wake properties of the two different rotor blades (straight and swept blades). The dynamics of this region is dominated by the blade characteristics (Ivanell et al., 2018). The near wake region is defined as one to five rotor diameters (D) distance away from the rotor disk (McKay et al., 2012). Thus, we have encompassed the near wake region with our measurements. In addition, in order to avoid laser scattering and reflection from the hub and rotor mount, and to ensure optimum quality of the acquired near wake data, we chose the distance to be $0.5D$ between the rotor plane and the measurement plane.

References:

- McKay, P., Carriveau, R., Ting, D. and Newson, T. (2012). Turbine wake dynamics. *Advances in Wind Power*, 65.
- Ivanell, S., Nilsson, K. and Eriksson, O. (2018). Wind turbine wakes and wind farm wakes. *Energiforsk Report 541*.

Response to Reviewer 2

1. How were both blades designed? Every blade necessitates the definition of the airfoil, the chord and the twist angle of each section. These information are not provided here. This paper would have been much more interesting if, given the same properties (airfoil, chord and twist angle), the swept blade had a different sweep angle only with respect to the standard (straight) blade (according to BEM theory, the two blades should be the same). I suspect this is not the case here, although this was not specified by the authors.

Answer: Thanks for the comment. We would like to emphasize that our main focus is on the bio-inspired blade whilst using a standard blade for comparison purposes solely. The general parameters for both blades (i.e., airfoil, chord length, span, twist angle, taper ratio, rotor diameter, etc.) were kept the same except the sweep angle. This yields a gematrical change in the splined leading and trailing edges present only in the swept model. In order to replicate the swift bird's wing (which is known as a high-performance flyer, capable of gliding for an extended period of time due to its wing characteristics (Lentink et al., 2007, Hedenström et al., 2016, Muir et al., 2017), the swept rotor blade was modelled by emulating the shape of the swift wing geometry as shown here in figure 1 (Fig. 1 in the revised manuscript). Also, both rotor blades operate at the same Reynolds number using the same rotor hubs. Under these flow conditions and blade configurations, we expect the differences in the near wake flow characteristics between the two blades to be associated with the sweep angle.

Figure 1: CAD model of A. Straight blade B. Swept blade

References:

- Lentink, D., Müller, U.K., Stamhuis, E.J., De Kat, R., van Gestel, W., Veldhuis, L.L.M., Henningson P. Hedenström, A., Videler J.J. and van Leeuwen, J.L. (2007). How swifts

control their glide performance with morphing wings. *Nature*, 446(7139), 1082-1085.

- Hedenström, A., Norevik, G., Warfvinge, K., Andersson, A., Bäckman, J. and Åkesson, S. (2016). Annual 10-month aerial life phase in the common swift *Apus apus*. *Current Biology*, 26(22), 3066-3070.
 - Muir, R.E., Arredondo-Galeana, A., and Viola, I.M. (2017). The leading-edge vortex of swift wing-shaped delta wings. *Royal Society Open Science*, 4(8), 170077.
2. It is very interesting that the ratio between the power estimated for the straight rotor P_w and the power of the free stream, P_f , is very close to the Betz limit (at least for the straight blade). For all the rotor model I have investigated, I have never exceeded a power coefficient of 0.4. Utility wind turbines oscillate between 0.43 - 0.5 depending on the accounted losses. Therefore, I cannot trust the estimate they made with P_w to be a reliable approximation of the actual power extracted by the turbine. Why was the actual power output not measured?

Answer: Thanks for the critical comment. It was a calculation mistake on our part. We did not include the power coefficient (C_p) in the power calculation. We revised our calculation in Table 1 including the power coefficient, which considers the effect of Reynolds number on the freely rotating rotor performance such as the rotational blades RPM, and aerodynamic forces (lift and drag) (Mingwei et al., 2016; Wilson et al., 1976).

Table 1: Wind turbine power characteristics

Case	U_∞ (ms^{-1})	Reynolds Number, $Re^{[1]}$	Tip Speed Ratio, $\lambda^{[2]}$		Power in the Freestream Flow, $P_f^{(3)}$ (mW)	Power Output of Wind Turbine, $P_w^{(4)}$ (mW)	
			Straight	Swept- back		Straight	Swept- back
1	0.12	23,800	2.53	1.92	27.14	9.30	6.83
2	0.15	29,700	3.28	2.10	53.01	18.20	11.82
3	0.18	35,700	3.78	2.21	91.61	31.53	18.00
4	0.21	41,600	4.34	2.30	145.47	50.25	25.04

$^1Re = \frac{U_\infty L}{\nu}$, where U_∞ is the freestream velocity, L is a characteristic length (rotor diameter), and ν is the kinematic viscosity of water.

$^2\lambda = \frac{\omega R}{U_\infty}$, Where ω is the turbine blade rotation per second, and R is the radius of wind turbine.

${}^3P_f = \frac{1}{2}\rho AU_\infty^3$, where ρ is the density of water, A is the swept area of the rotor (Johnson, 1985).

${}^4P_w = C_p \rho AU_1^2 (U_\infty - U_2)$, where, U_1 is the fluid velocity passing through the turbine rotor, U_2 is the mean velocity of the fluid downstream of the rotor. U_1 is estimated as: $U_1 = \frac{1}{2}(U_\infty + U_2)$. (Ragheb and Ragheb, 2011). C_p is the power coefficient and estimated as: $C_p = 0.593 - 0.565e$, where, $e = \frac{C_d}{C_l}$. (Mingwei et al., 2016; Wilson et al., 1976)

References

- Huleihil, M., & Mazor, G. (2012). Wind turbine power: The Betz limit and beyond. In *Advances in wind power*. IntechOpen.
 - Mingwei, G., Tian, D. and Deng, Y. (2016). Reynolds number effect on the optimization of a wind turbine blade for maximum aerodynamic efficiency. *Journal of Energy Engineering*, 142, 1.
 - Wilson, R.E., Lissaman, P.B., and Walker, S.N. (1976). *Aerodynamic performance of wind turbines*, Nasa Sti/recon Technical Report N, 77, 18598.
3. In a generic wind turbine, the thrust is very important as well. Here it was not measured but the authors estimated that with a 2-d estimate of the drag (not even axisymmetric, so quite wrongly balanced). One of my main objections is how we can compare these two different blades. One could compare two turbines with the same thrust coefficient or at least with the same power coefficient or tip-speed ratio. Right now, the two rotors share only the same free-stream velocity and the comparison is performed on things that are unrelated to the operating condition (such as the velocity deficit). For instance, consider the straight blade not rotating: the wake will be much weaker and according the analysis of the authors this case should be better in terms of wake deficit.

Answer: Thanks for the comment. We apologize for the confusion, but this study does not focus on wind turbine design but rather the effect of blade geometry on the near wake characteristics; in the same way that e.g., Tian et al. (2017) compared a wind turbine blade which was made out of long eared owl's airfoil with a NACA4412 based wind turbine blade. They compared their aerodynamic forces and power coefficients. Similarly, Aubrun et al. (2013) compared a non-rotating porous disk with a rotating turbine blade and their wake properties. These studies provided information about how flow properties change due to geometrical differences. In this study, similarly, we study near wake characteristics of two different freely rotating rotor blades. Based-on your comment, we understand that this was not communicated clearly enough. We have revised the title, abstract and manuscript text, hopefully to avoid any confusion.

We also agree with the reviewer that the drag curve in our study is not axisymmetric for the following reasons: First, the model was submerged in the flume. The flume dimensions were large enough so blockage ratio was not a concern; however, given the finite water height we cannot ignore some of the interactions that occurred with

the flume bottom boundary layer, which caused some deviation from axisymmetric. Second, and foremost, in the near wake flow field, mixing is somewhat restricted – resulting in asymmetries. In the far wake because of high mixing, the velocity deficit reflects a more gaussian curve (Garcia et al., 2017). As an example, below (figure 2), a deficit streamwise velocity profile from a wake analysis study (Aghsaee and Markfort, 2018) is shown, which depicts the measured profile to be asymmetric in the near wake. Asymmetric near wake velocity profiles have also been reported in Bartl et al. (2012).

Figure 2: Velocity deficit profiles at nine locations downstream of the turbine. Two dashed lines represent the top-tip and bottom-tip levels of the turbine. (Aghsaee and Markfort, 2018)

References:

- Tian, W., Yang, Z., Zhang, Q., Wang, J., Li, M., Ma, Y., and Cong, Q. (2017). Bionic design of wind turbine blade based on long-eared owl's airfoil. *Applied Bionics and Biomechanics*, 2017, 1–10.
- Aubrun, S., Loyer, S., Hancock, P.E., and Hayden, P. (2013). Wind turbine wake properties: Comparison between a non-rotating simplified wind turbine model and a rotating model. *Journal of Wind Engineering and Industrial Aerodynamics*, 120, 1–8
- García, L., Vatn, M., Mühle, F., and Sætran, L. (2017). Experiments in the wind turbine far wake for the evaluation of an analytical wake model. *Journal of Physics: Conference Series*, 854, 012015.
- Aghsaee, P. and Markfort, C.D. (2018). Effects of flow depth variations on the wake recovery behind a horizontal-axis hydrokinetic in-stream turbine. *Renewable Energy*, 125, 620-629.
- Bartl, J., Pierella, F., and Sætrana, L. (2012). Wake measurements behind an array of two model wind turbines. *Energy Procedia*, 24, 305–312.

4. The estimate of U_2 at page 5 relies too much in actuator disk theory. Real rotors create a turbulent wake, and the wake starts to recover back to the free-stream velocity so that the wake velocity is x dependent.

Answer: Thanks for the comment. In this study, U_2 (velocity at the wake) is estimated from the near wake velocity maps downstream of the rotor. The streamwise velocity profiles presented in figure 3 are calculated from the near wake velocity maps, which are both spatially and ensemble averaged over the rectangular PIV flow field domain (0.5D-2.1D along x). In this region, we do not expect any significant recovery to occur (Marten et al., 2020).

References:

- Marten, D., Paschereit, C.O., Huang, X., Meinke, M., Schröder, W., Müller, J., and Oberleithner, K. (2020). Predicting wind turbine wake breakdown using a free vortex wake code. *AIAA Journal*, 1-14.

Figure 3: Velocity deficit behind straight and swept rotor blade

5. Figure 3 should be normalized by the free-stream velocity as well (as done later on in figure 4). However, as the authors noted, the deficit is different, and I suspect this is due to the different thrust coefficients the turbines have. I think that a better comparison could have been to have the same thrust coefficient. check which one had the highest power coefficient. The latter requires an assessment of the angular velocity of the rotor and the torque and necessitates of another measurement technique (just monitoring the generator) or of stereo-PIV (if the most complicated way possible is chosen).

Answer: We thank the reviewer for this suggestion. We have normalized figure 3 (in manuscript) by the freestream velocity (see figure 4 below). The normalization is supported by the observation that the normalized velocity profiles in the wake behind

the classical blades collapse and present similar behavior across various Reynolds numbers. We have used the same approach for the profiles obtained for the swept blade for consistency and for the purposes of comparison. Indeed, the thrust coefficient is an important performance parameter for wind turbines, however, our goal was not to design a new wind turbine, rather to focus on the interactions formed between bio-inspired rotating blade and its wake.

Figure 4: Mean velocity vectors behind (A) straight blade, and (B) swept-back blade.

6. Figures 4, 5, 7 and 8 are normalized with the free-stream velocity. Now again I think that since the two rotors have different thrust coefficients, a fair comparison is not done by using the free-stream velocity as a reference scale. I think instead that the velocity field should be treated in terms of velocity deficit $U_1 - U(x, y)$ and normalized by the maximum velocity deficit measured somewhere (in agreement with a Galilean in-variance principle). This deficit velocity scale should be used to scale the turbulence statistics as well.

Answer: Thanks for the comment. We do not argue that the reviewer suggestion is invalid, we merely suggest a different approach. This different approach is used by other studies such as the one by Wu et al. (2019) where they study how the number of blades in wind turbine affect wake characteristics while normalizing their findings with free-stream velocity. Also, Aubrun et al. (2013) compared a non-rotating porous disk and a rotating turbine blade, and their wake properties were normalized with freestream velocity. In this study, both blades rotate freely due to incoming flow and their RPM depends on their tendency to rotate and generate enough torque based on

their aerodynamic characteristics. As the operating condition (Reynolds number) for both blades are similar and we kept the free stream velocity the same, it is plausible to assume that normalization should be performed using integral properties that characterize the flow condition studied here. Hence, we use the free stream as a reference value for comparing between the two cases

References:

- Wu, Y.T., Lin, C.Y., Huang, C.E., and Lyu, S.D. (2019). Investigation of multiblade wind-turbine wakes in turbulent boundary layer. *Journal of Energy Engineering*, 145(6), 04019023.
- Aubrun, S., Loyer, S., Hancock, P.E., and Hayden, P. (2013). Wind turbine wake properties: Comparison between a non-rotating simplified wind turbine model and a rotating model. *Journal of Wind Engineering and Industrial Aerodynamics*, 120, 1–8.

7. Page 7. The statement about the tip-speed ratio is new to me. I am aware of general momentum theory and Glauert optimum result, but if we consider freely rotating rotors, they rotate the fastest and do not produce anything.

Answer: We appreciate the reviewer's comment. As the reviewer mentioned, we looked for the statement about tip-speed ratio on page 7, but we did not find anything regarding TSR on page 7. We have thoroughly revised the text to avoid any further confusions.

8. Turbulence is not always negative. For wind farms, for instance, it provides a faster wake recovery and a shorter wake, enhancing the farm production. Less turbulence means longer wakes and lower power production, as observed for stable stratification conditions over wind farms.

Answer: Thanks for the comment. We agree with the reviewer's comment. We have revised the text as follows:

"Comparisons of turbulence profiles demonstrate that the swept blade exerts less turbulent stress into the wake flow compared to the straight blades at all Reynolds numbers. Higher turbulence increases load fluctuations on blades (Thomsen and Sorensen, 1999), which has negative effects on the performance of the blades to convert energy from the incoming flow. Swept rotor blades have been shown to reduce load fluctuations (Verelst and Larsen, 2010). However, the reduction of turbulence activity in the wake region sometimes may have an inverse effect as it can inhibit faster wake recovery and allow a longer wake signature in time and space (Ali et al., 2017)".

References:

- Thomsen, K., and Sørensen, P. (1999). Fatigue loads for wind turbines operating in wakes. *Journal of Wind Engineering and Industrial Aerodynamics*, 80, 121–136.

- Verelst, D.R., and Larsen, T.J. (2010). Load consequences when sweeping blades-a case study of a 5 MW pitch-controlled wind turbine. Risoe National Laboratory for Sustainable Energy, Denmark, Risoe-R-1724(EN).
 - Ali, N., Cortina, G., Hamilton, N., Calaf, M., & Cal, R. B. (2017). Turbulence characteristics of a thermally stratified wind turbine array boundary layer via proper orthogonal decomposition. *Journal of Fluid Mechanics*, 828,175-195
9. How is the circulation Γ estimated? This could be done by the instantaneous PIV velocity field but most likely it will just characterize the tip-vortex strength that is not necessarily the circulation at the blade (that by the way might not even be constant).

Answer: Thanks for the comment. The circulation is estimated using:

$$\Gamma = \iint_A \vec{\omega}_z \cdot d\vec{A} \quad (1)$$

where ω_z is the mean spanwise vorticity and A is the area of the near wake. Here, ω_z is defined as:

$$\omega_z = \frac{\partial v}{\partial x} - \frac{\partial u}{\partial y}$$

where u denotes the streamwise (x) velocity and v represents the normal (y) velocity. When the circulation is estimated in this plane (eqn. 1), this represents the circulation behind an airfoil (Stalnov et al., 2015; von-Karman and Sears, 1938).

References:

- Stalnov, O., Ben-Gida, H., Kirchhefer, A.J., Guglielmo, C.G., Kopp, G. A., Liberzon, A., and Gurka, R. (2015). On the estimation of time dependent lift of a European starling (*Sturnus vulgaris*) during flapping flight. *PLOS ONE*, 10(9), e0134582.
- von-Kármán, T. and Sears, W.R. (1938). Airfoil theory for non-uniform motion. *Journal of the Aeronautical Sciences*, 5(10), 379-390.